# IPA: Inference Pipeline Adaptation to Achieve High Accuracy and Cost-Efficiency

Saeid Ghafouri
*University of South Carolina*
*Queen Mary University of London*

Kamran Razavi
*Technical University of Darmstadt*

Mehran Salmani
*Technical University of Ilmenau*

Alireza Sanaee
*Queen Mary University of London*

Tania Lorido-Botran
*Roblox*

Lin Wang
*Paderborn University*

Joseph Doyle
*Queen Mary University of London*

Pooyan Jamshidi
*University of South Carolina*

## Abstract

Efficiently optimizing multi-model inference pipelines for fast, accurate, and cost-effective inference is a crucial challenge in machine learning production systems, given their tight end-to-end latency requirements. To simplify the exploration of the vast and intricate trade-off space of latency, accuracy, and cost in inference pipelines, providers frequently opt to consider one of them. However, the challenge lies in reconciling latency, accuracy, and cost trade-offs. To address this challenge and propose a solution to efficiently manage model variants in inference pipelines, we present IPA, an online deep learning **I**nference **P**ipeline **A**daptation system that efficiently leverages model variants for each deep learning task. Model variants are different versions of pre-trained models for the same deep learning task with variations in resource requirements, latency, and accuracy. IPA dynamically configures batch size, replication, and model variants to optimize accuracy, minimize costs, and meet user-defined latency Service Level Agreements (SLAs) using Integer Programming. It supports multi-objective settings for achieving different trade-offs between accuracy and cost objectives while remaining adaptable to varying workloads and dynamic traffic patterns. Navigating a wider variety of configurations allows IPA to achieve better trade-offs between cost and accuracy objectives compared to existing methods. Extensive experiments in a Kubernetes implementation with five real-world inference pipelines demonstrate that IPA improves end-to-end accuracy by up to 21% with a minimal cost increase. The code and data for replications are available at https://github.com/reconfigurable-ml-pipeline/ipa.

## 1 Introduction

Nowadays, companies run some or all of their Machine Learning (ML) pipelines on cloud computing platforms [70]. The efficient deployment of ML models is crucial in contemporary systems where ML inference services consume more than 90% of datacenter resources dedicated to ML workloads [12, 16]. In various critical applications, such as healthcare systems [29], recommendation systems [56], question-answering, and chatbots [20], a range of ML models, including computer vision models [29] and speech models [45], play an essential role. It is imperative to deploy these models cost-effectively while maintaining system performance and scalability.

Table 1: Comparison of IPA with previous works; Pipeline: A chain of models inference or just single model inference? Cost: Whether the work optimizes cost? Accuracy: Does it optimize accuracy? Adaptive: Can it adapt to different accuracy/cost optimization trade-offs?

| System | Pipeline | Cost | Accuracy | Adaptive |
|---|---|---|---|---|
| Rim [42] | ✓ | ✗ | ✓ | ✗ |
| INFaaS [60] | ✗ | ✓ | ✓ | ✗ |
| Inferline [25] | ✓ | ✓ | ✗ | ✗ |
| GPULet [22] | ✓ | ✓ | ✗ | ✗ |
| Llama [61] | ✓ | ✓ | ✗ | ✗ |
| FA2 [59] | ✓ | ✓ | ✗ | ✗ |
| Model Switch [75] | ✗ | ✗ | ✓ | ✗ |
| Scrooge [41] | ✓ | ✓ | ✗ | ✗ |
| Nexus [64] | ✓ | ✓ | ✗ | ✗ |
| Cocktail [36] | ✗ | ✓ | ✓ | ✗ |
| InfAdapter [63] | ✗ | ✓ | ✓ | ✓ |
| IPA | ✓ | ✓ | ✓ | ✓ |

Automatic resource allocation is a complex problem that requires careful consideration and has been extensively studied in various domains, including stream processing [27, 31, 46], serverless computing [52, 66], and microservices [30, 33, 76, 77]. Static auto-configuration of hardware resources [68], dynamic rightsizing of resources through autoscaling [73], and maximizing utilization with batching [13] are some of the techniques that have been used for resource management of ML models. In addition to efficient resource allocation, *accurate prediction* is another essential factor influencing ML model deployment. In many real-world scenarios, the predictions from these models have significant implications for business [9], industry [24], or human lives [1], and inaccuracies can lead to severe consequences [38, 49]. Hence, ensuring that ML models make accurate predictions is critical to producing reliable and trustworthy outcomes.

ML inference pipelines, as a chain of ML models, will raise several challenges for performance optimization. Unlike individually optimizing each stage, optimizing end-to-end ML inference pipelines will capture the correlation between configuration changes across multiple pipeline steps. Previous works [25, 41, 47, 61] have proposed solutions to address the challenges of efficient autoscaling, batching, and pipeline scheduling not only to consider the above challenges but also

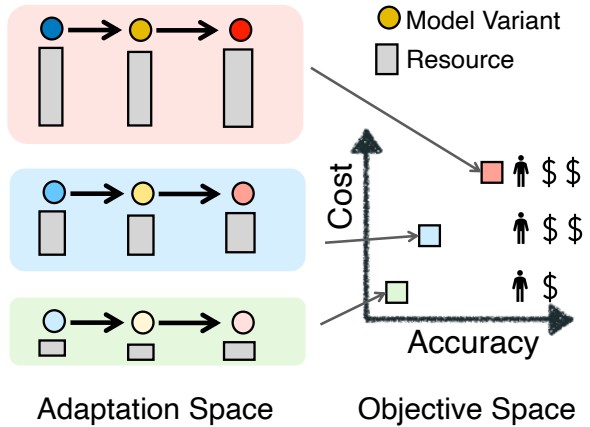

Figure 1: IPA provides a tunable framework for adjusting the system based on two contradictory cost and accuracy objectives.

to consider the dynamic nature of ML workloads. However, none of these approaches considers the combined optimization of accuracy and resource allocation across pipelines.

In ML inference pipelines, two adaptation techniques are commonly used: **Autoscaling**, which adjusts resources based on workload, and **Model-switching**, which employs different model variants with different accuracies/latencies to vary resource demands and tasks, allowing finer control over resource allocation and accuracy. The combination of these two techniques has been advocated to achieve more precise adjustments in accuracy and cost trade-offs. Previous autoscaling [62] and model-switching [63, 75] works have argued that using both techniques in conjunction with each other is beneficial in providing more precise adjustments in terms of accuracy and cost trade-offs, providing greater flexibility and efficiency in ML model resource allocation. However, none of the above works has considered optimizing accuracy and cost jointly in a multi-stage pipeline setting. Table 1 presents an overview of related inference serving works. Systems with inference *pipeline* serving have often overlooked the presence of multiple model variants for each inference task [25,41,47,59,61]. The heterogeneity of these model variants presents an opportunity not only to configure the pipeline to meet latency objectives but also to opportunistically select the most suitable model variant to enhance the accuracy of the pipeline output. On the other hand, previous works that have considered accuracy optimization of machine learning services [36, 60, 63, 75] do not support inference pipelines with an end-to-end optimization over all the stages of the inference pipeline.

In this paper, we propose IPA, a system for jointly optimizing accuracy and cost objectives. It can achieve multiple trade-offs between these two conflicting objectives based on the pipeline designer's preference. Figure 1 shows the premise of IPA that provides a tunable framework for adapting the

inference pipeline to achieve an optimal trade-off between accuracy and cost objectives given constraints and user preference. The choice of models in previous works was limited to using just one pre-selected model variant. IPA can broaden this search space by considering all model variants and dynamically adapting to a suitable choice of models based on the pipeline designer's preference.

The main contributions of this paper are as follows:

- We revisit the resource management design in inference pipelines by incorporating model switching, autoscaling, and pipeline reconfiguration (stage batch size). IPA proposes a new optimization formulation to allow for a more granular trade-off between the accuracy and cost objectives. It is also adaptable based on the inference pipeline designer's preference for each accuracy and cost objective.

- We propose a new optimization formulation based on the interaction between model switching, replication, and batch size. It can find (1) exact resources to allocate to each stage of the pipeline, (2) variants to decide for each stage, and (3) dependency between stages that enable accurate estimation of demand while guaranteeing end-to-end latency.

- The full implementation of IPA is built on top of Kubernetes. IPA integrates open-source technologies in its stack to facilitate seamless integration into production clusters.

- Experimental results show that IPA can achieve more granular trade-offs between the two contradictory objectives of cost and accuracy. In some scenarios, it was able to provide an improvement of up to 21% in the end-to-end pipeline accuracy metric with a negligible increase in cost.

## 2 Background and Motivation

This section provides the background of the inference pipeline and discusses the challenges of reconciling cost and accuracy trade-offs.

### 2.1 Search Space

Figure 2 illustrates the differences in latency and throughput between different versions of image classification models within the ResNet family. In particular, there exists an inverse relationship between **throughput** with both **latency** and **accuracy** for each variant of the model, considering the same number of **CPU cores** and fixed **batch sizes**. The variability in performance across model variants adds a new dimension to the configuration search space.

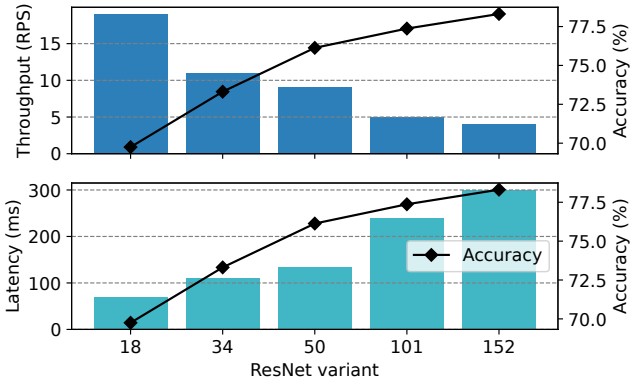

Figure 2: Performance difference across ResNet Family models for a batch size of one and one CPU core allocation.

Table 2: Performance difference across ResNet family models under different CPU allocations for a batch size of one, both blue and red core/model configuration can respond to 20 RPS and 75 ms throughput and latency requirements with different accuracy and costs.

| CPU Cores | ResNet18 | | ResNet50 | |
| --- | --- | --- | --- | --- |
| | Latency (ms) | Throughput (RPS) | Latency (ms) | Throughput (RPS) |
| 1 | 75 | 20 | 135 | 9 |
| 4 | 23 | 37 | 57 | 21 |
| 8 | 14 | 62 | 32 | 29 |

Choosing the best configuration among the highlighted parameters is non-trivial and subjective to multiple objectives and constraints, e.g., cost efficiency and Service Level Agreement (SLA) requirements between cloud users and providers. Table 2 shows that under the same 20 RPS incoming throughput and mutual SLA agreement of 75 ms between the user and the service provider, a user with high accuracy goals will choose a suitable configuration of four core assignments in ResNet50 (marked red). However, a user with lower accuracy demands will choose ResNet18 with one core, which can respond to latency and throughput requirements under lower core allocations (highlighted in blue).

## 2.2 Configuration Space

**Batch Size** Neural network structure provides parallel computation capability. Batching multiple requests will leverage the parallelization capability of neural networks and increase the utilization of assigned resources while increasing the total latency. Previous works [13, 25, 41] have shown that there is a relationship between the system utilization and the request latency, resulting in a non-trivial trade-off between maximizing the resource utilization without violating the latency SLA.

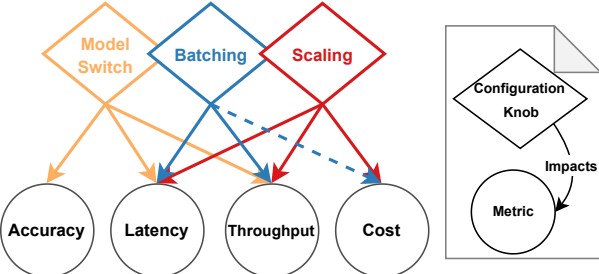

Figure 3: Impact of configuration knobs, batching indirectly affects the cost, e.g., decreasing the throughput will affect the IPA to more scaling and increase in the cost.

**Replication** There are mainly two resource provisioning techniques: vertical and horizontal scaling of resources. In vertical scaling, the assigned resources are modified, while in horizontal scaling, the number of replicas with the same amount of resources is adjusted to balance performance and cost. Horizontal scaling allows predictable performance using a similar environment [35], while vertical scaling allows a more fine-grained resource allocation to the ML model. We use horizontal scaling for the current work similar to [25, 59].

**Variant Selection** Previous works [60, 75] have shown that ML models are abundant for a single ML task. This brings the opportunity to abstract away the ML task from the underlying model and opportunistically switch the model based on the performance needs of the system.

Figure 3 shows the complex relationship between changing each configuration knob and performance objectives. Changing the batch size will affect the throughput and latency of each pipeline stage, while changes in the replication factor will directly impact the pipeline deployment cost. Model switching will result in changes both in accuracy and cost as different models have different resource requirements.

## 2.3 Inference Pipelines

Traditional ML applications revolve around the use of a singular deep neural network (DNN) to perform inference tasks, such as identifying objects or understanding natural language. On the contrary, modern ML systems (ML inference pipelines) are more intricate scenarios, such as digital assistant services like Amazon Alexa, where a series of interconnected/chained DNNs (in the form of DAG structures) are used to perform various inference tasks, ranging from speech recognition, question interpretation, question answering, and text-to-speech conversion, all of which contribute to satisfying user queries and requirements [59, 61]. As these systems frequently interact with users, it becomes essential to have a stringent SLA, which in our case is end-to-end latency.

The nonlinear dependency between the three configuration knobs (batch size, replication, and variant selection) and the pipeline variables introduces a complex decision space

between multiple conflicting goals.

Figure 6(a) depicts one of the evaluated pipelines consisting of two stages, an object detection stage, and an object classifier. A subset of the configuration space is presented in Table 3 with different batch sizes, model variants, and deployment resources. We denote cost as the number of replicas × allocated CPU cores per replica. The chosen configuration at each stage should first support the incoming workload into the pipeline while guaranteeing SLA, e.g., the sum of the latency of both stages should be less than the SLA requirement. Under the arrival rate of 20 RPS (Request Per Second) and the SLA requirement of 600 milliseconds, both combinations of (A1, B1) and (A2, B2) can meet the latency threshold and accommodate the incoming throughput. However, the first combination can support these requirements at the cost of $2 + 2 = 4$ CPU cores with a lower accuracy combination, while the latter can support the same load at the cost of $10 + 3 = 13$ CPU cores and a higher possible accuracy combination.

> **Challenge 1** Multiple configurations can satisfy the latency constraints of the inference pipeline. The "optimal" configuration depends on the accuracy and cost goals.

The next challenge is that in inference pipelines, the model selection at an earlier stage of the pipeline will affect the optimal model selection at downstream models, as the latency of a model at an earlier stage will affect the end-to-end latency. Consequently, the options available for the downstream models are more limited. In a similar example of pipeline Figure 6(a) and Table 3, and under an SLA of 500 ms, choosing a high latency and high accuracy configuration of A2 at the first stage will eliminate the variant B2 from the second stage's option.

> **Challenge 2** The choice of replication factor, batch size, and model variant is a joint decision in multiple stages of the inference pipeline.

## 3   System Design

This section provides a high-level overview of the main system components in IPA as illustrated in Figure 4.
**Model Loader** Users submit the models they intend to use for each stage of a pipeline. These models should provide a reasonable span of accuracy, latency, and resource footprint trade-offs.

Model variants can also be generated using model optimizers such as TensorRT [67], and ONNX graph optimization by using different quantization level of neural networks [32] and Neural Architectural Search methods [21]. After the model

Table 3: Two-stage pipeline tasks options. Latency is in Milliseconds and Cost is the number of physical cores.

| Variant | Scale | Batch | Latency | Cost | Accuracy |
|---|---|---|---|---|---|
| A1: YOLOv5n | 2 | 1 | 80 | $2 \times 1$ | 45.7 |
| A2: YOLOv5m | 5 | 1 | 347 | $5 \times 2$ | 64.1 |
| A3: YOLOv5n | 2 | 8 | 481 | $2 \times 1$ | 45.7 |
| A4: YOLOv5m | 5 | 8 | 1654 | $5 \times 2$ | 64.1 |
| B1: ResNet18 | 2 | 1 | 73 | $2 \times 1$ | 69.75 |
| B2: ResNet50 | 3 | 1 | 136 | $3 \times 1$ | 76.13 |
| B3: ResNet18 | 2 | 8 | 383 | $2 \times 1$ | 69.75 |
| B4: ResNet50 | 3 | 8 | 833 | $3 \times 1$ | 76.13 |

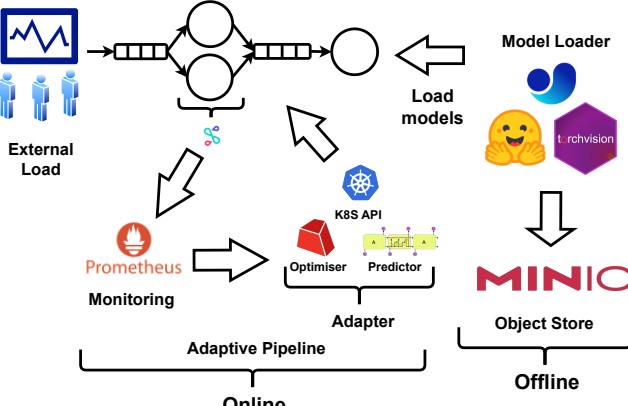

Figure 4: IPA system design. It consists of an offline phase for model profiling and an online phase for adaptive inference serving.

submission, the profiler (discussed in Section 4.2) will be executed for each model variant and store its latency under multiple batch sizes and resource assignments. Furthermore, as discussed in Section 4, the optimizer uses offline profiling to find the optimal solution during execution. Finally, the models are stored in object storage to reduce container creation times. In this work, we have used MinIO object storage [5].

**Pipeline System** After pipeline deployment through the model loader the pipeline is now ready to operate. Users will send inference requests to the pipeline to go through the models and return the inference predictions. Before models of each stage of the inference pipeline, a centralized load balancer distributes the inference requests between multiple stages of the inference pipeline. A centralized queue is behind each stage of the pipeline. Centralized queues help to have a deterministic queueing behavior and to efficiently model its latency, as described in Section 4. The queues of each stage then distribute the batched requests between model replicas. It uses a round-robin policy for load-balancing the batched requests between model replicas. Communications between multiple stages of the pipeline are implemented using gRPC [2]. The

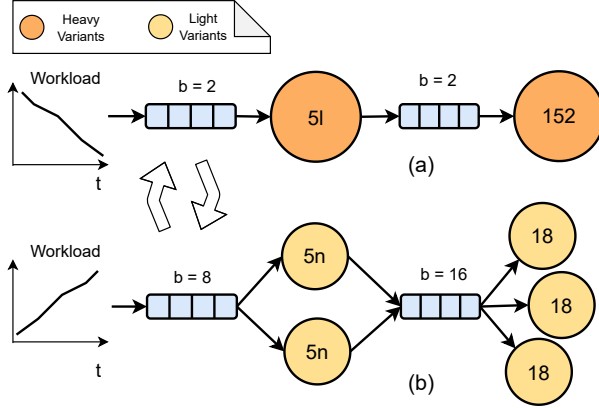

Figure 5: Switching between different configurations under (a) low and (b) high loads.

load balance between multiple containers of the same stage is achieved using Istio [3] sidecar containers. Each model container is deployed using Docker containers built from a forked version of MLServer [6] and Seldon Core [10] to implement the gRPC web servers and deployment on Kubernetes.

**Monitoring** The monitoring daemon uses the highly available time-series database Prometheus [8] underneath to observe incoming load to the system. It will periodically monitor the load destined for the pipeline.

**Predictor** In our load forecasting process, we employ an LSTM (Long Short-Term Memory), which is a type of recurrent neural network [40]. Our LSTM model is designed to predict the maximum workload for the next 20 seconds based on a time series of loads per second collected from the monitoring component over the past 2 minutes. To train the LSTM model, we utilized the initial two weeks of the Twitter trace dataset [15]. The architecture of our LSTM neural network consists of two layers, a 25-unit LSTM layer followed by a one-unit dense layer serving as the output layer.

**Runtime decisions for reconfigurations** The profiling data gathered by the profiler provide the latency and throughput of each model variant under different batch sizes. For runtime decision-making, a discrete event simulator uses these profiling data to estimate the end-to-end latency and throughput of the pipeline based on the number of replicas, model variants, and batch sizes at each stage. The predicted pipeline latencies and throughputs are then used by the optimizer in the adapter to determine the optimal configuration in terms of accuracy and cost objectives.

**Adapter** The Adapter is the auto-configuration module that periodically (1) fetches the incoming load from the monitoring daemon, (2) predicts the next reference load based on the observed historical load using the LSTM module, (3) obtains the optimal configuration in terms of the variant used, batch size, and number of replicas for the next timestep, and finally (4) the new configuration is applied to the pipeline through

Table 4: Notations

| Symbol | Description |
|---|---|
| $P$ | Inference pipeline |
| $s \in P$ | Inference pipeline stage |
| $SLA_s$ | Latency service-level agreement for each stage $s$ |
| $SLA_P$ | Latency service-level agreement for pipeline $P$ |
| $\lambda_P$ | Request arrival rate of pipeline $P$ |
| $M_s$ | Sets of available model variants for stage $s$ |
| $m$ | A model variant |
| $b_s$ | Batch size of stage $s$ |
| $R_m$ | Resource allocation of variant $m$ |
| $R_s$ | Resource allocation of stage $s$ |
| $a_m$ | Accuracy rank of variant $m$ |
| $n_s$ | Number of replicas of stage $s$ |
| $I_{s,m}$ | Indicator of activeness of variant $m$ in stage $s$ |
| $PAS$ | Pipeline accuracy score |
| $q_s(b_s)$ | Queuing time of stage $s$ under batch size $b$ |
| $l_{s,m}(b_s)$ | Latency of variant $m$ under batch size $b_s$ |
| $h_{s,m}(b_s)$ | Throughput of variant $m$ under batch size $b_s$ |
| $A_P$ | End-to-end accuracy of pipeline $P$ |
| $th$ | Threshold RPS of base allocation of $R_m$ |
| $\alpha$ | Accuracy objective weight |
| $\beta$ | Resource allocation objective weight |
| $\delta$ | Penalty term for batching |

the Kubernetes Python API [4].

Figure 5 shows a snapshot of the system with two available options per model in a video analysis pipeline; the first options are $\{YOLO5l, YOLO5n\}$ and the second are $\{ResNet18, ResNet152\}$. In low loads (a), choosing more accurate models $YOLO5l$ and $ResNet152$ with small batch sizes is preferable to ensure low latency. However, in higher loads, it is preferable to choose lightweight models such as $YOLO5n$ and $ResNet18$ with more replication and larger batch sizes to ensure high throughput for the system.

## 4 Problem Formulation

We now present the details of the optimizer discussed in Section 3. Problem formulation requires a robust definition of the accuracy of the inference pipeline and offline latency profiles of the model variants. Sections 4.1 and 4.2 discuss details of the inference pipeline accuracy definition and profiling methodology, respectively.

### 4.1 Accuracy Definition over Pipeline

To the best of our knowledge, there is only a direct way to compute the end-to-end accuracy of a pipeline if one evaluates the accuracy of the entire pipeline on the labeled data designed for the pipeline. However, for inference pipelines with stages having different semantics (e.g., a speech model connected to a language model) and multiple model options for each stage,

finding the accuracy of all the options by curating handpicked datasets for each pipeline composition is not feasible. In this work, we used a heuristic to rank the inference pipelines. We have only considered linear inference pipelines with one input and one output stage where a set of consecutive models are connected. The accuracy of each model is computed offline and is part of the model's property.

In this work, we do not consider model drifts [53], therefore, we use the per-stage statically computed accuracies to compute the end-to-end inference pipeline accuracy. For models with a qualitative performance measure other than accuracy (e.g., mAP for object detection, 1-WER for speech recognition tasks, and ROUGE for NLP tasks), as long as we have the specification *higher (or lower) means better* in the measure, we can substitute accuracy with that measure. To define a metric over the accuracy in a pipeline, we use independent stage accuracy to compute the end-to-end accuracy over the entire inference pipeline. With the assumption of independence of errors between different stages of the inference pipeline, we have used *multiplying* of each stage of the inference pipeline accuracy as a heuristic to evaluate the overall accuracy preference between different combinations of models in the inference pipeline. We will refer to this metric for inference pipeline accuracy as **P**ipeline **A**ccuracy **S**core abbreviated as *PAS* metric.

The end-to-end accuracy is typically influenced by the combination of errors at each stage, and the relationship between these errors may not be accurately captured by multiplication. If errors are correlated or there are dependencies between stages, this method might deteriorate the soundness of the proposed inference pipeline's accuracy. However, we argue that through the evaluation of alternative accuracy metrics presented in Appendix C, it becomes evident that until this problem is fully addressed by the machine learning community, the multiplication of individual stage accuracies serves as the most practical measure for approximating end-to-end accuracy.

## 4.2 Profiler

The formulation of the joint cost-accuracy problem needs information on the latency, accuracy, and throughput of each model variant. Previous works have discovered that [35,41,59, 60] the latency of a model under a specific resource allocation is predictable based on the incoming batch sizes. The profiler will record the latency on the target hardware for different batch sizes under specific allocations of resources. We found in our experiments that assigning memory beyond a specific value to the models does not impact performance; therefore, we only need to find enough memory allocation for each node, which will be the memory requirement for running the largest batch size.

Finding a minimum allocation to containers is necessary since we have chosen horizontal scaling for workload adap-

Table 5: Sample CPU cores base allocation for different YOLO variants under different RPS thresholds (Capped on maximum 32 cores).

| load | YOLOv5n | YOLOv5s | YOLO5m | YOLOv5l | YOLO5x |
|------|---------|---------|--------|---------|--------|
| 5 | 1 | 1 | 4 | 8 | 16 |
| 10 | 1 | 2 | 8 | 16 | ✕ |
| 15 | 1 | 8 | 16 | 32 | ✕ |

tation in this work. Previous works on CPU evaluation for inference graphs [47,59] have assigned one core to each container. Adapting a similar approach is not practical in our case, as more resource-intensive models cannot execute inference under given latency requirements with one core per container. Therefore, we find a base allocation for each model variant regarding the number of cores that can provide a reasonable base performance. The solver selects the model variants and horizontally scales them with the chosen base allocations.

Following previous works [34,59], we define the per-stage latency $SLA_s$ as the average latency of all available variants for the task to serve batch size one under the base resource allocation multiplied by 5 as suggested by Swayam [34]. The pipeline $SLA_P$ that is later used in Section 4.3 is defined as the sum of per-stage $SLA_s$, ($SLA_P = \sum_{s \in P} SLA_s$). Table 5 provides some of the base CPU allocations under the 5 RPS, 10 RPS, and 15 RPS thresholds for five variants (YOLO5n, YOLO5s, YOLO5m, YOLO5l, and YOLO5x) of the object detection stage. The values in the allocation columns show the minimum number of CPU allocations per container needed to respond to a certain load in the stage $SLA_s$. We refer to these values as the base resource allocation to a model variant. The allocation of base resources for all stages ($\forall s \in S$) and their corresponding model variants ($\forall m \in M_s$) is the minimum number of resources in terms of CPU cores (Equation 1a) that can respond to a certain threshold (Equation 1b) and met the predefined base latency requirements per stage $SLA_s$ for the largest batch size in our system (Equation 1c). Therefore, the minimum number of resources per model variant can be formulated as follows:

$$\min R_m \qquad (1a)$$
$$\text{subject to } th \leq h(m, R_m) \qquad (1b)$$
$$l_m(\max(b_s)) \leq SLA_s, \qquad (1c)$$

The value *th* in Equation 1b is a hyperparameter of the IPA solver, and its values are empirically found for each pipeline. The hyperparameters used in IPA are reported in Appendix A. In summary, Equation 1a is used to find the base allocation for each model variant where the threshold *th* is fixed and $R_m$ is the resource requirements of model variant *m*. Therefore, the base resource allocation for all models can be found statistically. In the same Object Detector task with different model variants, as shown in Table 5, we choose

the first configuration row, which supports the highest RPS with the minimum resource allocation per container. As a consequence, the allocation of base resources is fixed during runtime, and the performance of each stage of the pipeline $h(m, R_m)$ will be a function of used model variant $m$, and its throughput $h(m)$, and the number of replicas $n_s$.

For profiling, we follow [60] and record latency and throughput on the power of two increments of 1 to 64 batch sizes. Profiling per all batch sizes is costly; therefore, following [59] for each model variant, we fit the observed results on the profiled batch sizes under the base resource allocation to a quadratic polynomial function $l_m(b_s) = \alpha b_s^2 + \beta b_s + \gamma$ that can infer the latency for unmeasured batch sizes with a lower Mean Squared Error (MSE) than a linear function, $\alpha b_s + \beta$. Multiple model variants are available per-stage of the inference pipelines, and the mentioned approach can decrease the profiling cost by an order of magnitude.

## 4.3   Optimization Formulation

We used Integer Programming (IP) as a robust optimization framework to address the intricate challenges associated with configuring inference pipelines. As we have discussed later in Section 4.4, due to the optimality of the IP results compared to heuristic solutions, we have chosen IP modeling alongside the Gurobi solver to guarantee the optimality of the results. We have discussed the scalability and limitations of the IP formulation and Gurobi solver in Section 5.3. The goal of IPA is to maximize accuracy and minimize cost while guaranteeing SLAs.

$$\text{Objectives} = \begin{cases} \text{Maximizing the Accuracy} \\ \text{Minimizing the Cost} \end{cases} \quad (2)$$

There are $|M_s|$ model variants for each inference stage $s \in P$ in the pipeline $P$. Each $m \in M_s$ model variant performs the same inference tasks (e.g., image classification) that exhibit different resource requirements, latency, throughput, and accuracy. The resource requirements of the models are estimated offline in the profiling step (Section 4.2). The profiler provides the resource requirements of the model variants per task and their latencies under different batch sizes. The two main goals of IPA are maximizing the precision of the pipelines and minimizing the resource cost using lighter models. We define $I_{s,m}$ as an indicator of whether a selected model variant is currently active for task $s$ or not:

$$I_{s,m} = \begin{cases} 1 & \text{if } m \text{ is active in stage } s \\ 0 & \text{Otherwise} \end{cases} \quad (3)$$

At each point in time, only one model variant $m$ can be active for each inference stage; therefore, the resource requirement of each stage model replicas $R_s$ is equal to that stage active variant resource requirement $R_m$.

$$R_s = \sum_{m \in M_s} R_m \cdot I_{s,m} \quad (4)$$

Similarly, the latency and throughput of each pipeline stage ($l_s$ and $h_s$) are calculated based on the latency and throughput of the active model variant for that stage.

$$l_s = \sum_{m \in M_s} l_{s,m}(b_s) \cdot I_{s,m} \quad (5)$$

$$h_s = \sum_{m \in M_s} h_{s,m}(b_s) \cdot I_{s,m} \quad (6)$$

Another contributing factor to the pipeline's end-to-end latency is the time spent on the queue of each inference stage. For queue modeling, we have used the theoretical upper bound formulation introduced in [59]:

$$q_s(b_s) = \frac{b_s - 1}{\lambda} \quad (7)$$

Equation 7 illustrates the worst-case queuing delay based on the arrival rate and the batch size. The first request that arrives in a batch should wait for $b_s - 1$ additional request before being sent to the models.

The formal definition of the metric *PAS* (explained in Section 4.1) for the accuracy of the end-to-end pipeline is defined in Equation 8. *PAS* is computed by multiplying the accuracies of the active models in each stage of the inference pipeline.

$$PAS = \prod_{s \in P} \left( \sum_{m \in M_s} a_{s,m} \cdot I_{s,m} \right) \quad (8)$$

The multi-objective goal (9) is to maximize the end-to-end accuracy of the pipeline and minimize the cost. The cost objective is achieved in two ways, using smaller models (models with fewer resource requirements) and using the least number of replicas for them. The batch size for each model should be chosen carefully, as larger batch sizes will increase the utilization and throughput of the entire load and the per batch latency. It is worth mentioning that the IPA multi-objective formulation is agnostic to inference pipeline accuracy definition (as also shown later in Appendix C). The existing end-to-end inference graph accuracy definition can be substituted with potentially better future inference pipeline accuracy definitions.

$$\begin{aligned} f(n, s, I) = &\ \alpha \cdot PAS \\ &- \beta \sum_{s \in P} n_s \cdot R_s \\ &- \delta \sum_{s \in P} b_s \end{aligned} \quad (9)$$

The two variables $\alpha$ and $\beta$ adjust the preference level given to each objective. We should find a batch that increases utilization on a reasonable scale. Following [59], we have added

a small penalty term for batch size in the objective function $\delta$ that tries to minimize batch sizes until lowering batch sizes causes instability in the system (the system throughput becomes lower than the incoming workload). We now can describe the auto-configuration of the three online configuration knobs explained in Section 2.2 as an IP problem:

$$\max \quad f(n, s, I) \tag{10a}$$

$$\text{subject to} \quad \sum_{s \in P} l_s(b_s) + q_s(b_s) \leq SLA_P, \tag{10b}$$

$$\text{if } I_{s,m} = 1, \text{ then}$$

$$n_s \cdot h_s(b_s) \geq \lambda_p, \quad \forall s \in P \tag{10c}$$

$$\sum_{m \in M_s} I_{s,m} = 1, \quad \forall s \in P \tag{10d}$$

$$n_s, b_s \in \mathbb{Z}^+, \quad I_{s,m} \in \{0, 1\}, \quad \forall s \in S, \forall m \in M_s \tag{10e}$$

The chosen combination of models should be able to meet the latency (10b) constraint of the pipeline. The pipeline $SLA_P$ is the aggregate of per-stage $SLA_s$ as described in Section 4.2. The end-to-end latency of the pipeline is obtained by summing the inference latency of the chosen model variant $l_s(b_s)$ and the queueing time of each model server $q_s(b_s)$ in the inference path. Also, the sum of the throughput of all replicas of an active model should be higher than the arrival rate 10c into the pipeline.

In summary, the objective function tries to find the most accurate combination of models in the inference pipeline while allocating the least number of physical resources based on the pipeline designer's preference. This trade-off between the two objectives is configurable by modifying the $\alpha$ and $\beta$ weights for accuracy and resource allocation. Therefore, the inputs of the optimization formulation are resource requirements $R$, latency $l$, precision of all models $a$, and throughputs $h$ of all model variants $m$ in all stages $s$ under all possible batch sizes $b$ alongside the queueing latency model $q$ under all batch sizes, the incoming load $\lambda$ coming to the pipeline and pipeline latency $SLA_p$. The output returns the optimal number of replicas $n$, batch size $b$, and the chosen model variant $I$ for each pipeline stage.

## 4.4 Gurobi Solver

IPA has been designed to meet production-level requirements, (1) guaranteeing an optimal solution, (2) no need for additional pretraining or retraining costs, and (3) negligible overhead. Other approaches like heuristics are ad hoc solutions that are hard to generalize to different systems, and ML approaches do not guarantee the optimal solution (failing to find the optimal solution results in over-provisioning or under-provisioning) and typically incur long training times (for example, reinforcement learning). On the contrary, although the IP formulation is NP-hard (as shown in [60]), it meets

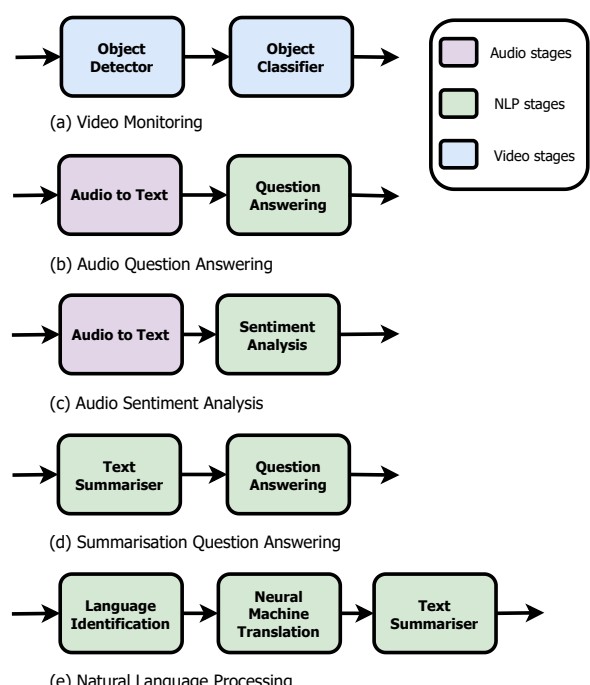

Figure 6: Representative pipelines used in this work.

the above requirements for a production-ready system. In our case, we chose the Gurobi solver [19] that guarantees the optimal solution; the downside of using the solvers is that in the case of very large search spaces or under very tight SLA requirements, it might take a long time to find the desirable configuration. In our case, Gurobi solved the problem formulation in Formula 10 in less than a second.

## 4.5 Dropping

High workloads may cause high back pressure in the upstream queues. If a request has already passed its *SLA* at any stage in the inference pipeline, then there might be no point in continuing to serve it until the last stage and placing high pressure on the system. A mechanism we have used is to drop a request at any stage of the pipeline if it has already passed *SLA* in the previous steps. We also consider that a request is dropped if its current latency exceeds 2× the SLA to avoid constant back pressure on the queues.

## 5 Evaluation

In this section, we conduct extensive experiments to demonstrate the practical efficacy of IPA in real-world scenarios using a diverse set of workloads. The code and data for replicating the results are available at https://github.com/reconfigurable-ml-pipeline/ipa.

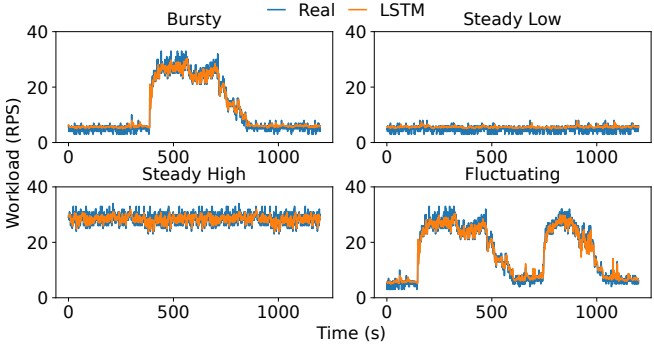

Figure 7: Representative tested load patterns from the Twitter trace [15], showing LSTM predictions.

Table 6: per-stage and end-to-end SLA of inference pipelines (in seconds).

| Pipelines | Stage 1 | Stage 2 | Stage 3 | E2E |
|---|---|---|---|---|
| Video Monitoring | 4.62 | 2.27 | ✗ | 6.89 |
| Audio QA | 8.34 | 0.89 | ✗ | 9.23 |
| Audio Sentiment | 8.34 | 1.08 | ✗ | 9.42 |
| Sum QA | 2.52 | 1.32 | ✗ | 3.84 |
| NLP | 0.97 | 12.76 | 3.87 | 17.61 |

## 5.1 Experimental Setup

Most previous works on inference pipelines [18, 25, 58, 61] have implemented the entire pipeline on their self-made infrastructures. We have implemented our frameworks on top of Kubernetes, the de facto standard in the containerized world and widely used in industry. This will enable easier access to the framework for future use by developers. IPA is implemented in Python with over 8K lines of code, including the adapter, simulator, queuing, load balancer, and model container implementations.

**Hardware** We used six physical machines from Chameleon Cloud [48] to evaluate IPA. Each server is equipped with 96 Intel(R) Xeon(R) Gold 6240R CPU @ 2.40GHz cores and 188 Gb of RAM.

**Pipelines** We used five descriptive pipelines with a wide variety of models for each stage, as shown in Figure 6. The pipelines are adapted from previous work and also from industrial examples. The video monitoring pipeline (pipeline a) is a commonly used pipeline in previous works [25, 74] and industry [23] which an object detector sends the cropped images to a later model to perform classification tasks like detecting a license plate or human recognition. The audio and question answering / sentence analysis pipelines (pipelines b and c) are adapted from use cases that comprise multiple types of ML model [28]. NLP pipelines (pipelines d and e) are representative examples of emerging use cases of language models [71, 72]. For a full specification of the models used in each stage of the pipelines, refer to Appendix A. Also, for

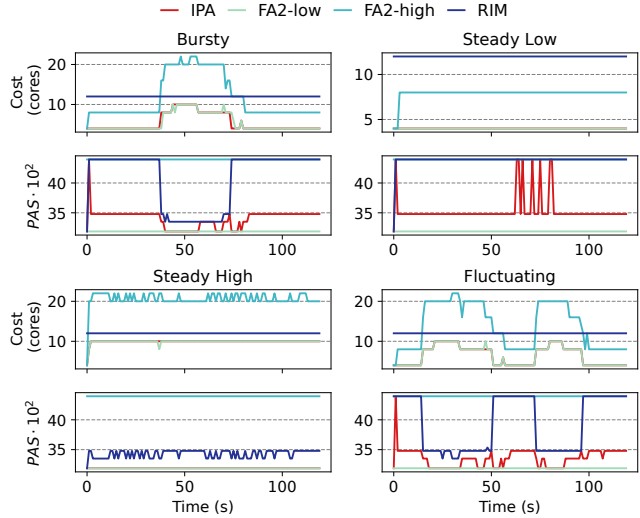

(a) Temporal analysis under different workloads.

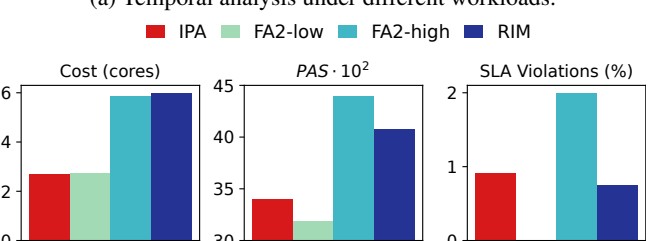

(b) Average analysis on bursty workload

Figure 8: Performance analysis of the Video pipeline.

the $\alpha$, $\beta$ and $\delta$ values of Equation 9 used for the experiments IPA, see Appendix B.

**Baselines** We compare IPA with variations of two similar systems, namely FA2 [59] and RIM [42]. FA2 is a recent system that achieves cost efficiency using scaling and batching; however, compared to IPA, it does not have model switching as an optimization angle. RIM, on the other hand, does not have scaling as a configuration knob but uses model switching to adapt to dynamic workloads. The original RIM does not include batching; we also add batching to RIM for a fair comparison. As RIM does not support scaling, we statically set the scaling of each stage of the inference pipeline to a high value. Similarly, FA2 does not support model switching; therefore, we use two versions of it, one FA2-low, which sets the model variants to the lightest models, and FA2-high, which sets the model variants to a heavy combination of models on each stage [1]. The three systems compared benefit from the LSTM predictor explained in Section 3.

**Workload** Figure 7 shows excerpts from Twitter trace [15] that have been used to evaluate the performance of IPA against

---

[1]Ideally we should have set the FA2-high to the heaviest models but due to resource limitations, we set it to models that on average give better accuracy compared to IPA

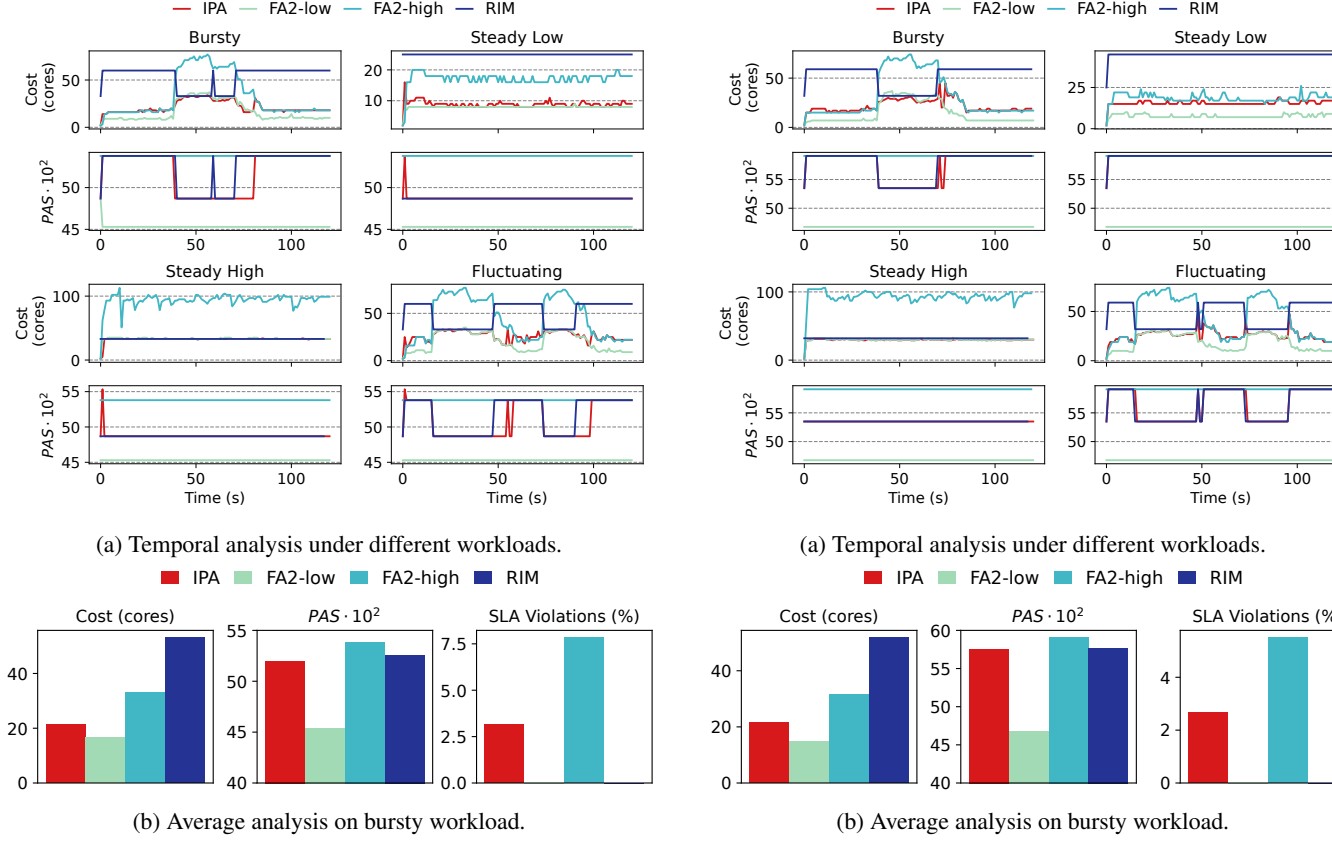

(a) Temporal analysis under different workloads.

(b) Average analysis on bursty workload.

Figure 9: Performance analysis of the Audio-qa pipeline.

(a) Temporal analysis under different workloads.

(b) Average analysis on bursty workload.

Figure 10: Performance analysis of the Audio-sent pipeline.

four parts of the data set. It includes bursty, fluctuating, steady low, and steady high workloads. For the train and test split during the training of the LSTM module, we trained the LSTM module on 14 days of the Twitter trace and chose the four mentioned workload excerpts from the other 7 unseen parts of the dataset. The LSTM predictor can predict the workload in less than 50*ms* with a Symmetric Mean Absolute Percentage Error (SMAPE) [39] of 6.6% that is comparable to the predictors used in systems with similar context [78]. Furthermore, an asynchronous load tester was implemented to emulate the behavior of users in real-world data centers.

**SLA** Table 6 shows the pipeline SLA of each inference pipeline that is calculated by summing the heuristic per-stage SLAs explained in Section 4.2.

## 5.2 End-to-End Evaluation

Figure 8 shows the evaluation results for the video pipeline in the four bursty, steady high, steady low, and fluctuating workloads. Since FA2-high and FA2-low are always set to the lightest and heaviest variants, they will always provide the lowest and highest possible *PAS* despite load fluctuations. In the three bursty, steady low, and fluctuating workloads IPA can always achieve a trade-off between the cost objective and,

in steady high workload IPA, diverge to a configuration that uses the lowest-cost model variants to adapt to the high resource demands of steady high workload. The only available adaptation mechanism for RIM is to change the models; therefore, under load variations in bursty and fluctuating workload, it trades off accuracy for meeting the incoming load burst throughput. Failing to meet the incoming workload will result in request drops and SLA violations. As expected, FA2-low and FA2-high have the highest and lowest SLA attainment. In the video pipeline, IPA provides the same resource efficiency as FA2-low, as the base allocation (explained in Section 4) of variants used for the first stage in the video pipelines is similar in most cases, and changing the model in favor of latency reduction does not result in higher computational costs (this is the reason for the overlap between FA2-low and IPA in the cost temporal plots shown in Figure 8a.). In total, IPA can show a better balance between the two cost and accuracy objectives. Although FA2-high and RIM provide the highest accuracies, their cost efficiency is compromised due to employing more accurate variants per stage and a high scaling factor. FA2-low can meet the requirements of SLA and achieve the same cost efficiency as IPA but cannot improve accuracy because it is fixed in the lightest variants. The higher violation rate in FA2-high, RIM, and IPA compared

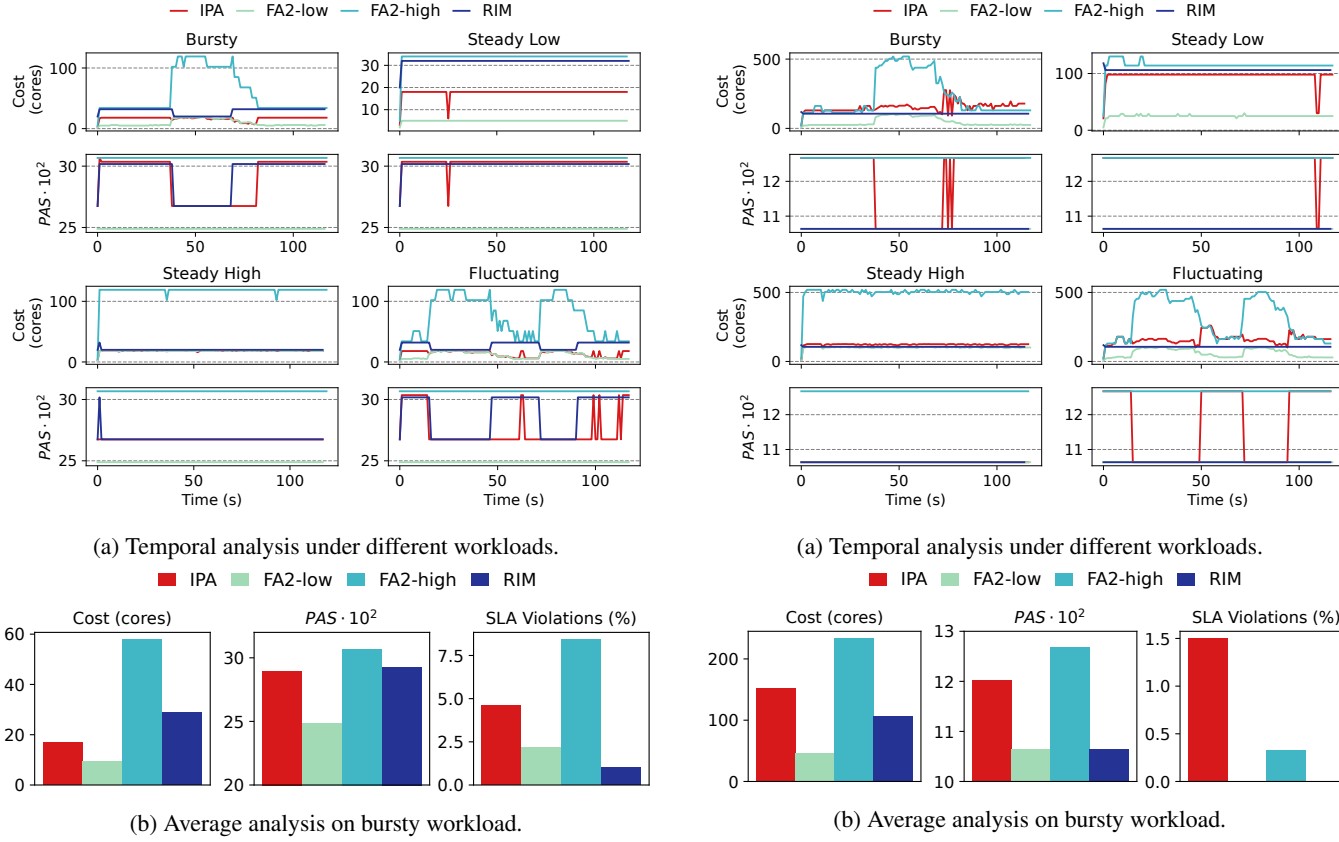

(a) Temporal analysis under different workloads.

(b) Average analysis on bursty workload.

Figure 11: Performance analysis of the Sum-qa pipeline.

(a) Temporal analysis under different workloads.

(b) Average analysis on bursty workload.

Figure 12: Performance analysis of the NLP pipeline.

to FA2-low is because SLAs are defined using the processing latency of average models, and SLAs become tighter for more accurate models, resulting in a higher tail latency violation. Figures 9a and 10a show the same temporal and average results on the audio-qa and audio-sent inference pipelines. Due to the lower number of variants used in these two pipelines ($5 \times 5 = 25$ for video and $5 \times 2$ and $5 \times 3$ in the audio-qa and audio-sent pipelines), we observe fewer fluctuations in RIM in all workloads. However, like the video pipeline, IPA achieved a trade-off between the accuracy and cost objectives.

Compared to the stages in the three mentioned pipelines, the base allocations for the summarization stage used in the sum-qa and NLP pipelines provide a larger span of changes in the required CPU cores (see Appendix A). For example, the resource difference between the heaviest and lightest model in the Object Detection stage of the video pipeline is $8 - 1 = 7$, while it is more than doubled in the summarization stage ($16 - 1 = 15$). Consequently, we observe a longer span of differences between the FA2-low and FA2-high approaches in these two pipelines. In both approaches, IPA can adapt to the load using the second least heavy models, resulting in a cost reduction of 3x and 2x with only 2 and 1 loss value in the inference *PAS*. It should be noted that the initial spikes in *PAS* are due to the initial setting. The optimizer then immediately

adjusts the models in response to the workload.

## 5.3 IPA Scalability

**System Scalability** To examine the effectiveness of IPA in real-world systems, we included the NLP pipeline in our evaluations, which during bursts scales to more than 500 cores (Figure 12a). Due to the use of best production-grade ML deployment practices, such as lightweight containers and Kubernetes as the back-end with the benefit of distributed scheduling and cluster management, we believe IPA has the potential to scale to large clusters.

**Optimizer Scalability** As mentioned in Section 4, we have used the Gurobi solver to solve the IP optimization problem. One critique of using Gurobi is its limitations in the solvable problem space in the time constraints of a real-world autoscaler. To guarantee fast autoscaler adaptation to workload fluctuations, the autoscaler should be able to find the next configuration in less than two seconds to leave enough room for the adaptation process itself, which in our experiments was around the same number of eight seconds that sums up $8 + 2 = 10$ which we used as our adaptation monitoring interval. We conducted a set of simulated experiments shown in Figure 13 to examine the decision-making time of the IPA

Figure 13: Decision time of Gurobi optimizer for IPA formulation with respect to the number of models and tasks on the inference graph.

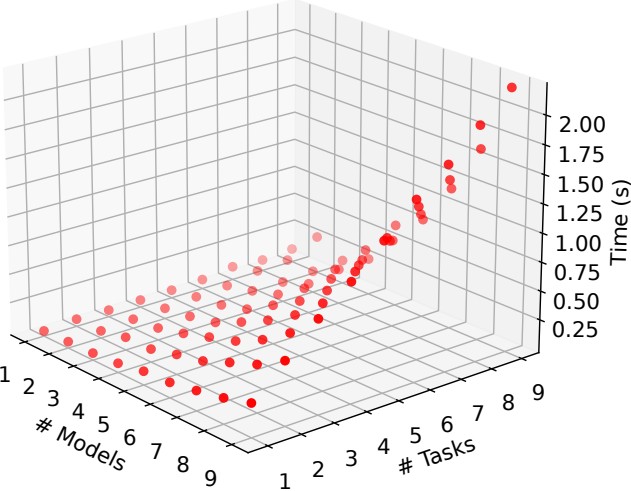

Figure 14: Comparison of IPA results for different trade-offs between accuracy and cost objectives, IPA can navigate effectively between the two cost and accuracy objectives.

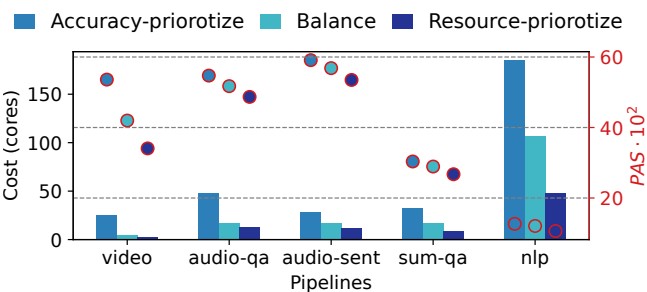

growth with respect to the change in the number of available model variants and the number of tasks in the inference pipelines (length of the inference graph). IPA can find the optimal configuration for inference pipelines with 10 stages, each with 10 models, in less than 2 seconds. Having an effective decision time for inference pipelines beyond these sizes requires faster optimization solutions. In addition, IPA will adapt to workloads with more SLA violations for pipelines that have SLA requirements lower than the decision-making time of IPA (although it will eventually adapt). However, most existing inference pipelines [57] rarely go beyond 10 stages; therefore, IPA will be effective for real-world use cases.

## 5.4 IPA Adaptability

The main premise of IPA is to provide an adaptable framework to achieve a trade-off between the cost and accuracy objectives

utilizing the three configuration knobs of model switching, scaling, and batching. Instead of using a fixed value for $\alpha$ and $\beta$ in previous experiments, we examined the effect of changing the weights given to each objective by modifying the values of $\alpha$ and $\beta$ for each inference pipeline. Figure 14 shows a set of experiments carried out on all five pipelines, where in one scenario, cost optimization (resource prioritize) is set as a priority by setting $\beta$ to a higher value, and in another scenario, accuracy is set as a system priority by using a higher value for $\alpha$. It is evident that IPA provides an adaptable approach to optimize different cost and accuracy preferences by the inference pipeline designer. For example, a highly accurate adaptation scenario can be chosen for the audio-sent pipeline with 28 CPU cores and an average *PAS* of 59 or a lower accurate result of 53 with 11 CPU cores.

Figure 15 shows the end-to-end latency CDF of the five pipelines tested to further show the flexibility of IPA in dynamic workloads. IPA leverages its fast adaptation by using heavy models only when the load is low and achieves nearly the same latency efficiency as FA2-low (with a higher accuracy than FA2). Only RIM can provide better latency compared to IPA, but as shown in the previous examples (e.g., Figure 9b for the Audio-qa pipeline) comes at the expense of high resource allocations (3x compared to IPA in the same pipeline).

## 5.5 IPA Predictor

Predictors are effective in reducing SLA violations. Most previous work on inference pipeline serving [25, 41, 42, 59] has done reactive auto-configuration. In reactive approaches, configuration changes occur with live load monitoring and in response to load changes. [73, 78] for predicting load before load changes. IPA uses a proactive approach using an LSTM predictor that takes advantage of historical data. The ablation analysis provided in Figures 16 shows that using the LSTM predictor is beneficial in reducing SLA violations in all pipelines with a negligible difference in resource consumption. The LSTM module is trained in less than ten minutes for the 14 days of Twitter traces; therefore, using it is practical in real-world scenarios. Furthermore, comparing with the baseline, which is a predictor that has complete knowledge of the load in the future interval (ground truth of the LSTM predictor), shows that in the case of video, audio-qa and audio-sent pipelines, the IPA predictor performs well with the baseline predictor. Moreover, the baseline predictor results in sum-nlp and the nlp results show that better load predictions in the prediction module can also potentially result in further reduction in SLA.

## 6 Related Works

**Single stage inference serving**: Several approaches have been proposed in previous research to improve performance

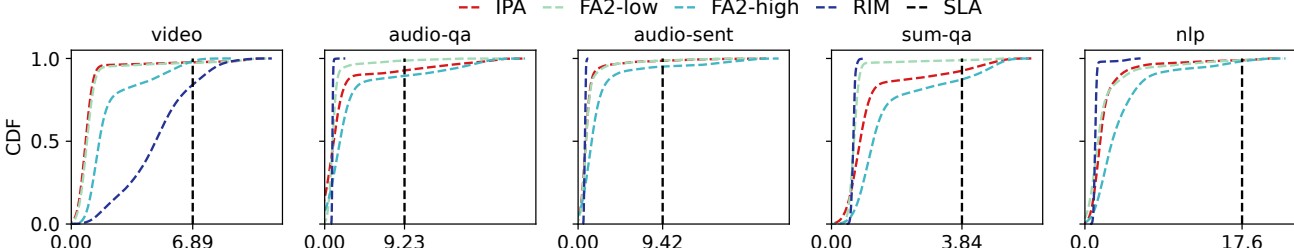

Figure 15: End-to-end latency distribution for the five tested inference pipelines under different approaches. IPA can achieve latency close to the FA2-low with light model variants, and only RIM is achieving better latency at the expense of high resource over-provisioning.

Figure 16: Effect of using predictor on reducing SLA violations on bursty workload, IPA LSTM can reduce SLA violations up to 10x with the same resource usage. Also, using baseline predictors with perfect knowledge about the future reveals potential room for reduction of SLA with better predictors.

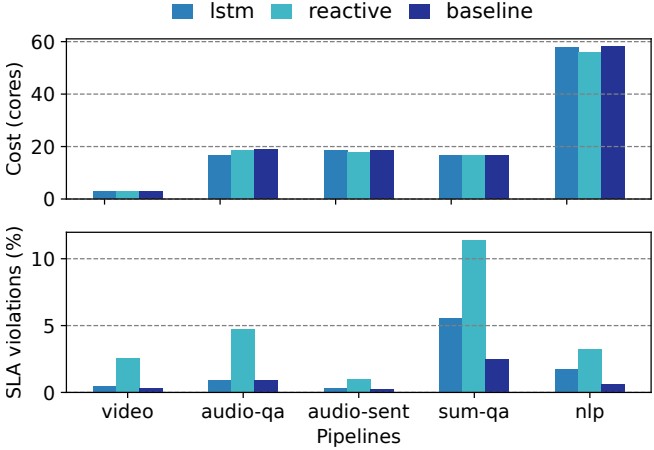

model variants in the single-stage setting from the user and automatically selects the best-performing model within the user-defined SLOs. It also actively loads and unloads models based on their usage frequency. InfAdapter [63] and Cocktail [36] propose joint optimization formulations to maximize accuracy and minimize cost with predictive autoscaling in single-stage inference scenarios.

**Multi-stage inference serving**: Several approaches have been proposed in previous research to improve inference performance metrics in multistage inference serving systems [7, 25, 37, 41, 42, 44, 47, 51, 58, 59, 61, 65, 69] since changing one model's configuration affects subsequent steps. InferLine [25] reduces the end-to-end latency of ML service delivery by heuristically optimizing configurations such as batch sizes and horizontal scaling of each stage. Llama [61] is a pipeline configuration system specific to the use case designed exclusively for video inference systems. It attempts to reduce end-to-end latency by interactively decreasing the latency of each stage in the pipeline. Stages of the pipeline can be either ML inference or non-ML video tasks like decoding. GrandSLAm [47] is a system designed to minimize latency and ensure compliance with SLA requirements in the context of a chain of microservices mainly dedicated to ML tasks. The system achieves this by dynamically reordering incoming requests, prioritizing those with minimal computational overhead, and batching them to maximize each stage's throughput. FA2 [59] provides a graph transformation and dynamic programming solution in inference pipelines with shared models. The graph transformation part breaks the execution graph to make it solvable in real-time, and the dynamic programming solution returns the optimal batch size and scaling factor per each DNN stage of the pipeline. Multimodel and Multitask inference VR/AR/Metaverse pipelines [17, 50] are other emerging use cases of inference pipelines. However, unlike IPA, none of the above approaches considers the three pillars of accuracy, cost, and end-to-end latency/throughput jointly for multistage inference serving systems.

metrics without considering multiple stage inference models [13, 26, 34, 68, 73]. They intend to enhance a set of performance metrics, e.g., latency and throughput, and reduce resource utilization through adaptive batching, horizontal and vertical resource scaling, model switching, queue reordering, and efficient scheduling of models in heterogeneous clusters. A few works have considered joint optimization of qualitative metrics like accuracy and performance for single-stage inference-serving systems. Model Switching [75] proposes a quality adaptive framework for image processing applications. It switches between models trained for the same task configuration knob and switches from heavier models to lighter models in response to load spikes. However, the proposed prototype does not consider the interaction of model switching between other resource configuration knobs, such as autoscaling and batching. INFaaS [60] abstracts the selection of

# 7 Conclusion and Future Works

In this work, we introduced IPA, an online auto-configuration system for jointly improving the resource cost and accuracy over inference pipelines. IPA uses a combination of offline profiling with online optimization to find the suitable model variant, replication factor, and batch sizes for each step of the inference pipeline. Real-world implementation of IPA and experiments using real-world traces showed that it could preserve the same cost efficiency and SLA agreement while also having an improvement of up to 21% in the proposed end-to-end pipeline accuracy metric (*PAS*) compared to the two baseline approaches. The following are some directions for future work.

**Scalability.** IPA leverages Gurobi solver for finding the suitable configurations. This worked fine in our problem setting, as a limited number of model variants were used in each step of the pipeline. However, an interesting future direction for IPA is to examine its performance where more model variants are available for each step of the pipeline and also in cases where we have more complicated larger graphs [7]. The adapter needs to be able to respond to bursts in less than one second, which demands either designing new heuristic methods that can find a good enough but not necessarily optimal solution or data-driven solutions like some of the methods that have been used before in similar auto-configuration context like Bayesian Optimization [14], Reinforcement [68] Learning or Causal Methods [43].

**Emerging ML deployment paradigms.** Multi-model serving [12] enables more efficient usage of GPUs. This feature enables the simultaneous loading of several models on the same server instead of running one microservice per model, as in IPA. Although the focus of IPA was on the CPU service, using it on GPUs and containerized platforms is not straightforward. To our knowledge, there is no built-in mechanism for sharing GPUs on mainstream container orchestration frameworks like Kubernetes. Making the IPA formulation consistent with the sharing of GPUs and also considering interference between multiple models on the scheduler [55] as part of the IPA is a potential future extension.

## Acknowledgements

This work has been supported, in part, by the National Science Foundation (Awards 2007202, 2107463, 2038080, and 2233873), Deutsche Forschungsgemeinschaft (DFG, German Research Foundation) – Project-ID 210487104 - SFB 1053, and Roblox Corporation. The authors also thank Chameleon Cloud for providing cloud resources for the experiments.

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

# Appendix

# A    Pipelines Stages Specifications

List of models used per stage of the pipeline with their specifications. BA in the following tables refers to the Base Allocation explained in Section 4.2 in terms of CPU cores.

**Object Detection**
Performance Measure: Mean Average Precision (mAP)
Number of Variants: Five
Source: Ultralytics YOLOV5 [11]
Threshold: 4 RPS

Table 7: Object Detection Task Models

| Model | Params (M) | BA | mAP |
|-------|-----------|----|----|
| YOLOv5n | 1.9 | 1 | 45.7 |
| YOLOv5s | 7.2 | 1 | 56.8 |
| YOLOv5m | 21.2 | 2 | 64.1 |
| YOLOv5l | 46.5 | 4 | 67.3 |
| YOLOv5x | 86.7 | 8 | 68.9 |

**Object Classification**
Performance Measure: Accuracy
Number of Variants: 5
Source: Torchvision [54]
Threshold: 4 RPS

Table 8: Object Classification Task Models

| Model | Params (M) | BA | Accuracy |
|-------|-----------|----|---------|
| ResNet18 | 11.7 | 1 | 69.75 |
| ResNet34 | 21.8 | 1 | 73.31 |
| ResNet50 | 25.5 | 1 | 76.13 |
| ResNet101 | 44.54 | 1 | 77.37 |
| ResNet152 | 60.2 | 2 | 78.31 |

**Audio**
Performance Measure: Word Error Rate (WER) [2]
Number of Variants: Five
Source: HuggingFace
HuggingFace Source: facebook
Threshold: 1 RPS

---

[2]WER is the primary metric for evaluating audio-to-text models, however, to preserve the higher means better property mentioned in Section 4.1 we used the 1-WER which is similar to the accuracy measure for other types of model.

Table 9: Audio Task Models

| Model | Params (M) | BA | 1 - WER |
|---|---|---|---|
| s2t-small-librispeech | 29.5 | 1 | 58.72 |
| s2t-medium-librispeech | 71.2 | 2 | 64.88 |
| wav2vec2-base | 94.4 | 2 | 66.15 |
| s2t-large-librispeech | 267.8 | 4 | 66.74 |
| wav2vec2-large | 315.5 | 8 | 72.35 |

**Question Answering**
Performance Measure: F1 Score
Number of Variants: Two
Source: HuggingFace
HuggingFace Source: depeest
Threshold: 1 RPS

Table 10: Question Answering Task Models

| Model | Params (M) | BA | F1 Score |
|---|---|---|---|
| roberta-base | 277.45 | 1 | 77.14 |
| roberta-large | 558.8 | 1 | 83.79 |

**Summarisation**
Performance Measure: Recall-Oriented Understudy for Gisting Evaluation (ROUGE-L)
Number of Variants: Six
Source: HuggingFace
HuggingFace Source: sshleifer
Threshold: 5 RPS

Table 11: Summarisation Task Models

| Model | Params (M) | BA | ROUGE-L |
|---|---|---|---|
| distilbart-1-1 | 82.9 | 1 | 32.26 |
| distilbart-12-1 | 221.5 | 2 | 33.37 |
| distilbart-6-6 | 229.9 | 4 | 35.73 |
| distilbart-12-3 | 255.1 | 8 | 36.39 |
| distilbart-9-6 | 267.7 | 8 | 36.61 |
| distilbart-12-6 | 305.5 | 16 | 36.99 |

**Sentiment Analysis**
Performance Measure: Accuracy
Number of Variants: Three
Source: HuggingFace
HuggingFace Source: Souvikcmsa
Threshold: 1 RPS

Table 12: Sentiment Analysis Task Models

| Model | Params (M) | BA | Accuracy |
|---|---|---|---|
| DistillBerT | 66.9 | 1 | 79.6 |
| Bert | 109.4 | 1 | 79.9 |
| Roberta | 355.3 | 1 | 83 |

**Language Identification Task Models**
Performance Measure: Accuracy
Number of Variants: One
Source: HuggingFace
HuggingFace Source: dinalzein
Threshold: 4 RPS

Table 13: Language Identification Task Models

| Model | Params (M) | BA | Accuracy |
|---|---|---|---|
| roberta-base-finetuned | 278 | 1 | 79.62 |

**Neural Machine Translation**
Performance Measure: Bilingual Evaluation Understudy (BELU)
Number of Variants: Two
Source: HuggingFace
HuggingFace Source: Helsinki-NLP
Threshold: 4 RPS

Table 14: Neural Machine Translation Task Models

| Model | Params (M) | BA | Accuracy |
|---|---|---|---|
| opus-mt-fr-en | 74.6 | 4 | 33.1 |
| opus-mt-tc-big-fr-en | 230.6 | 8 | 34.4 |

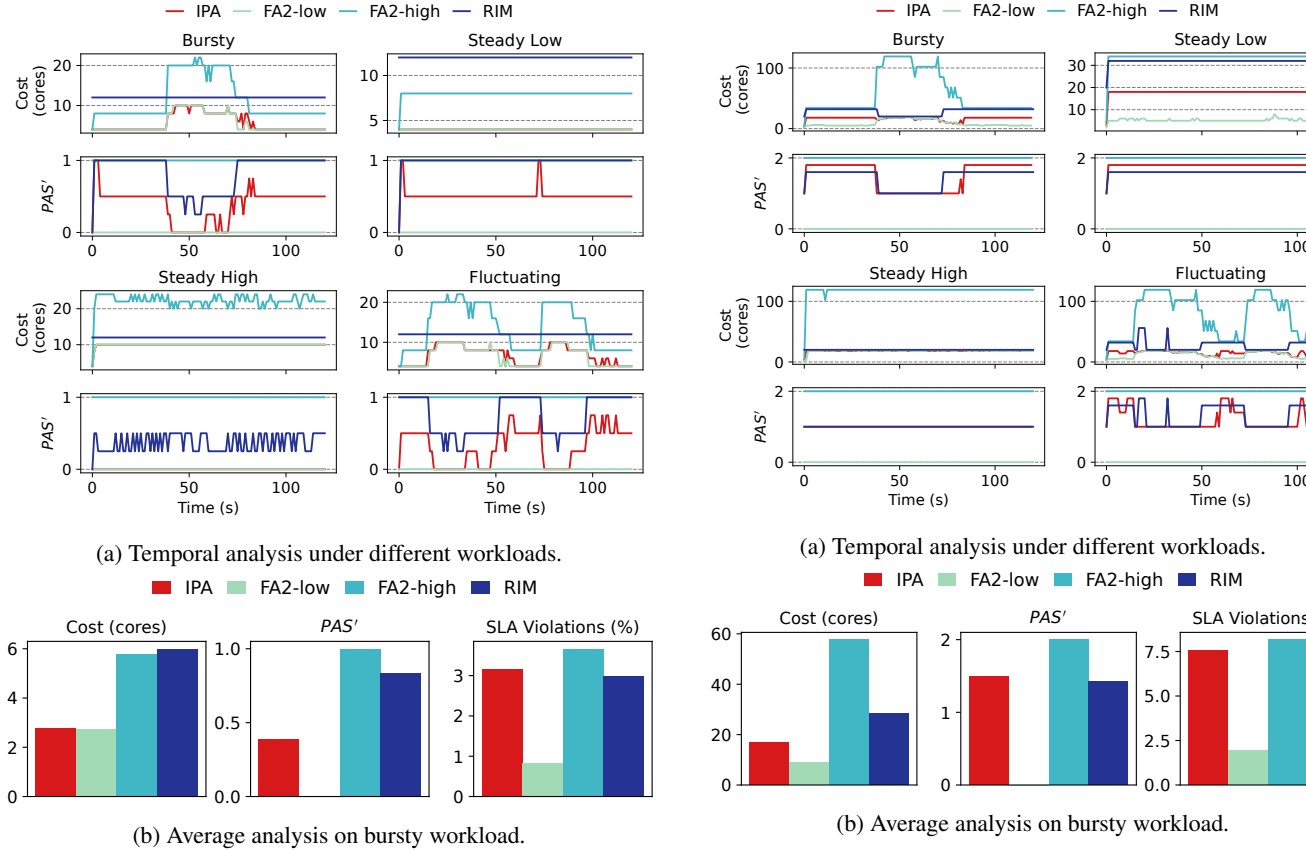

(a) Temporal analysis under different workloads.

(b) Average analysis on bursty workload.

Figure 17: Performance analysis of the Video pipeline.

(a) Temporal analysis under different workloads.

(b) Average analysis on bursty workload.

Figure 18: Performance analysis of the Sum-qa pipeline.

## B    Constant Multipliers Values

In this part, we have presented the values used for the multiplier α, β, and δ in Equation 9. These values were applied in the experiments discussed in Section 5.2. It is important to mention that because various objectives (like accuracy and cost) have different scales, the multiplier's scale is adjusted accordingly. We empirically found the best values for each objective multiplier.

Table 15: Pipelines objectives multiplier values

| Pipeline | α | β | δ |
| --- | --- | --- | --- |
| Video | 2 | 1 | 0.000001 |
| Audio-qa | 10 | 0.5 | 0.000001 |
| Audio-sent | 30 | 0.5 | 0.000001 |
| Sum-qa | 10 | 0.5 | 0.000001 |
| NLP | 40 | 0.5 | 0.000001 |

## C    Alternate Inference Pipeline Accuracy Definition

In this section, we have presented results for an alternative accuracy metric in inference pipelines that we will refer to as $PAS'$. Our goal was to empirically show that two different accuracy metrics were able to provide similar results. To define a metric over the accuracy in a pipeline, we first sort the accuracy of each stage's model variants from lowest to highest. Then, we assign a zero scale to the least accurate model and one to the most accurate model (normalizing the accuracy). Intermediate model variants are assigned scaled accuracy values between zero and one, proportionally aligned with their rankings in the ordered list. For example, if three model variants exist, the model scaled accuracy is assigned 0, 0.5, and 1. The overall accuracy over the pipeline is considered as the sum of each stage's model variants' scaled accuracy value. For example, a two-stage pipeline with three model variants per stage and the second most accurate model chosen in each pipeline will have an end-to-end accuracy rank of 0.5 + 0.5 = 1. As a consequence, this will also change Equation 8 for the accuracy objective to sum up the normalized accuracy at each step instead of multiplication. Also, the accuracy values of each model and stage $a_{s,m}$ are the nor-

malized accuracy explained rather than the actual accuracy values. This transforms Equation 8 into Equation 11. This *PAS′* replaces the *PAS* metric in Equation 9. The rest of the IP formulation presented in Section 4 will remain the same.

$$PAS' = \sum_{s \in P} \big( \sum_{m \in M_s} a_{s,m} \cdot I_{s,m} \big) \qquad (11)$$

We replicated the end-to-end experiments outlined in Section 5.1 by substituting the accuracy metric with a new metric across all pipelines. In all cases, this alternative metric exhibited the same trend of the multiplication heuristic results across different methods. Figures 17 and 18 present the results obtained using this novel accuracy metric. A comparison with the results in Section 5.2, specifically Figures 8 and 11, reveals that both sets of results align in terms of resource allocation and accuracy optimization.

