# OpenReview forum: "IPA: Inference Pipeline Adaptation to Achieve High Accuracy and Cost-Efficiency"
_JSYS/2024/March_Papers — JSYS 2024 March Papers Submission_

### Official Review · Reviewer_1X8y · 2024-03-15
**Review of revised manuscript**

**Decision:**

Strong accept: excellent paper that will help the community

**Review:**

- as before -

**Strengths:**

- as before -

**Weaknesses:**

The two main weaknesses I had pointed out have been resolved.

**Detailed Comments:**

- As before but all nits have been addresses. Thanks!

**Expertise:**

Follow the literature closely, last published 5+ years ago

**Summary Of Review:**

The authors have completed the two major revisions I had asked for to my satisfaction.

- I am pleased to see the new end to end accuracy estimate as a product of accuracies of individual stages. This is far more natural and does not have the degenerate case of a zero accuracy pipeline stage like before. I appreciate also the incorporation into the ILP problem in Eq (8).

- Happy with the rewrite of Sec 4.3 in particular the fixes to symbols and cleanup of the writing, it is a lot more clear now.

**Usefulness:**

Yes

---

### Official Review · Reviewer_rgdC · 2024-04-13
**Review of revised manuscript**

**Decision:**

Strong accept: excellent paper that will help the community

**Review:**

As before.

**Strengths:**

As before.

**Weaknesses:**

Most weaknesses have been resolved.

**Expertise:**

Actively publishing in this area

**Summary Of Review:**

The authors have provided a reasonable feedback to the raised concerns and have limited the scope of the work  to workload requests per second but not data drifts. It would be helpful if the authors include a discussion section that provides justification for discounting transition costs in the current formulation and which realistic settings/SLA requirements may necessitate consideration of transition costs.

Otherwise, happy to see a revised better version.

**Usefulness:**

Yes

---

### Official Review · Reviewer_gwFH · 2024-04-13
**Review of Paper Revision**

**Decision:**

Strong accept: excellent paper that will help the community

**Review:**

as before -

**Strengths:**

as before -

**Weaknesses:**

Most have been addressed.

**Detailed Comments:**

as before -
Thank you for the effort in addressing the comments.

**Expertise:**

Follow the literature closely, last published 5+ years ago

**Summary Of Review:**

It's good to see that authors take into account the feedback provided in previous reviews and incorporate it into their work. The reviewed version not only demonstrates significant improvement but also highlights and motivates the research contribution of the work.

**Usefulness:**

Yes

---

### Author Response · Authors · 2024-04-19
**Final Version Submitted**

Dear JSys Program Chair,

We have updated the pdf with the camera-ready version, please let us know if you have any questions.

With kind regards

---

> ### Author Response · Authors · 2024-04-19
>
> There was an update on one of the author's affiliations that we corrected it in the camera-ready version, please use the latest uploaded pdf as the reference for the publication.

---

### Meta-Review · Program_Chairs · 2024-04-15

**Recommendation:** Accept
**Confidence:** 5

**Metareview:**

On behalf on the Journal of Systems Research, we are pleased to accept your paper. Congratulations!

You have one month to produce the final camera-ready version. You can upload the final version earlier once the changes are done.

Please follow the instructions here: https://www.jsys.org/instructions#preparing-the-final-version

---

### Decision · Program_Chairs · 2024-04-15

Accept (with shepherding)